# Optical Response Characteristics of Single-Walled Carbon Nanotube Chirality Exposed to Oxidants with Different Oxidizing Power

**DOI:** 10.3390/molecules26041091

**Published:** 2021-02-19

**Authors:** Yuji Matsukawa, Kazuo Umemura

**Affiliations:** Department of Physics, Graduate School of Science, Tokyo University of Science, 1-3 Kagurazaka, Shinjuku, Tokyo 162-8601, Japan; meicun2006@163.com

**Keywords:** carbon nanotube, DNA, chirality, oxidizing power, near-infrared, redox

## Abstract

Semiconductor single-walled carbon nanotubes (SWNTs) have unique characteristics owing to differences in the three-dimensional structure (chirality) expressed by the chiral index (n,m), and many studies on the redox characteristics of chirality have been reported. In this study, we investigated the relationship between the chirality of SWNTs and the oxidizing power of oxidants by measuring the near-infrared (NIR) absorption spectra of two double-stranded DNA-SWNT complexes with the addition of three oxidants with different oxidizing powers. A dispersion was prepared by mixing 0.5 mg of SWNT powder with 1 mg/mL of DNA solution. Different concentrations of hydrogen peroxide (H_2_O_2_), potassium hexachloroidylate (IV) (K_2_IrCl_6_), or potassium permanganate (KMnO_4_) were added to the dispersion to induce oxidation. Thereafter, a catechin solution was added to observe if the absorbance of the oxidized dispersion was restored by the reducing action of the catechin. We found that the difference in the oxidizing power had a significant effect on the detection sensitivity of the chiralities of the SWNTs. Furthermore, we revealed a detectable range of oxidants with different oxidizing powers for each chirality.

## 1. Introduction

Single-walled carbon nanotubes (SWNTs) are versatile nanomaterials with many notable electronic and mechanical properties. They are expected to be integral to the realization of a low-carbon society. As “green molecules”, they are also expected to cause breakthroughs in the fields of biotechnology and energy supply systems. Unfortunately, the hydrophobic nature of SWNTs severely limits their chemical, biochemical, and biological applications. To separate the SWNT bundles, SWNT powder has been mixed with double-stranded DNA (dsDNA) and sonicated under appropriate conditions to form a dsDNA-SWNT complex in which the dsDNA molecules wrapped the SWNT surface [1,2,3,4,5,6,7,8,9].

SWNTs have a structure in which a graphene sheet is rolled into a cylindrical shape, and they typically have a diameter of few nanometers. To form a seamless cylindrical tube, it is necessary to take two of the hexagons in a graphene lattice and overlap them. A vector connecting the centers of the two hexagons is represented by the chiral index (n,m) (chirality), which denote the number of unit vectors along two directions in the crystal lattice of graphene sheet. The chiral index determines the structure of a single-walled carbon nanotube. Qin reported that the chiral index of SWNTs can be determined accurately using nano-beam electron diffraction [10]. Recently, the responsiveness of ten chiralities of SWNTs by increasing exposure time and measuring photoluminescence has been reported [11].

SWNTs exhibit properties of metals and semiconductors by the difference in the chirality. Semiconductor SWNTs have a unique diameter according to their chirality, and the bandgap energy changes because of the unique band structure [12,13,14,15,16,17,18,19,20,21]. Many studies have been conducted to apply the chirality-dependent redox characteristics to biosensors [22,23,24,25,26,27,28,29,30,31,32,33,34].

Knorr et al. measured the photoluminescence (PL) intensity of sodium dodecyl sulfate (SDS)-SWNT complexes and investigated the correlation between the response characteristics of six chiralities [(8,3), (7,5), (10,2), (7,6), (12,1), (10,3)] and the change in oxidant (hypochlorite [NaOCl] or hydrogen peroxide [H_2_O_2_]) concentration. In their study, they indicated that the degree of quenching was higher for NaOCl than for H_2_O_2_ in all chiralities, even though the molar concentration of the latter was higher than that of the former (approximately 9 × 10^−6^ M NaOCl vs. 7.9 × 10^−5^ M H_2_O_2_). This result suggested that NaOCl is a stronger oxidizing agent than H_2_O_2_ [35]. Weisman et al. changed the pH by adding H_2_SO_4_ to SWNTs suspended in SDS and monitored the correlation between the pH and fluorescence intensity. When the pH was gradually reduced from 8.0, the emission intensities were quenched in order of the chiralities: (7,5), (8,3), and (6,5). This phenomenon showed a difference in the sensitivity of chirality to oxidation [36]. Hamano et al. investigated the optical response sensitivity of SWNT hybrids prepared by mixing dsDNA and carboxymethyl cellulose (CMC). In that study, SWNT hybrids with different mixing ratios were oxidized with H_2_O_2_ followed by reduction with catechin, and the absorption spectra were measured. They detected the optical properties of the SWNTs by focusing on the (8,4)/(9,4)/(7,6) and (10,5)/(8,7) chiralities with large absorbance changes. They showed that the optical response of SWNTs is highly dependent on the mixing ratio of dsDNA and CMC [37]. Ishibashi et al. detected the antioxidant capacity of Japanese tea and catechins by measuring both absorbance and PL. They also investigated the effect of pH on the sensitivity of oxidation and reduction detection, showing that the (9,4) chirality was the most sensitive to the catechin reduction reaction at an emission wavelength of 1000–1400 nm and a pH of 8.0 [38].

Zheng et al. found that electron transfer occurs readily between small-molecule redox reagents and semiconducting carbon nanotubes. They identified a direct correlation between the bandgap of semiconductor nanotubes and their reduction potential, showing that (6,5)-enriched SWNTs can be easily oxidized with potassium (IV) chloroiridate (K_2_IrCl_6_) as the oxidant, and they found that changes in the concentration affected the absorption spectra [39]. However, they did not report on the KMnO_4_-induced redox action of SWNTs.

To our knowledge, no studies have considered the difference in the oxidizing powers of oxidants to investigate the oxidation properties of SWNT chirality. In the current study, we focused on the detection sensitivity of the chirality of SWNTs and oxidizing powers of oxidants, measured the absorption spectra using oxidants with different oxidizing powers, and investigated the effect of the difference in oxidizing power on the chirality. The suspension was reduced with catechin after oxidation, and we verified the absorbance was recovered by reduction. The oxidizing powers of the oxidizing agents used in this study were −1 for H_2_O_2_, +4 for the Ir of K_2_IrCl_6_, and +7 for the Mn of KMnO_4_. We also specified the detectable concentration range of the oxidizing agents for each chirality.

From these results, we clarified that the change in absorbance differs depending on the chirality, even at the same oxidant concentration. We found that the detection sensitivity of chirality to detect oxidants depends on the oxidizing power. In other words, the (6,5) chirality is suitable for oxidants with a high oxidizing power, and the (8,7) or (9,4) chirality is suitable for detecting the concentration of oxidants with a low oxidizing power.

## 2. Results

Figure 1 shows a conceptual diagram of the experiment. The dsDNA-(6,5)-enriched SWNT and dsDNA-HiPco SWNT suspensions were oxidized with H_2_O_2_, K_2_IrCl_6_, or KMnO_4_ and then reduced with catechin to detect changes in the NIR-absorbance spectra.

Figure 2a–c show examples of the changes in the NIR-absorbance spectra for the dsDNA-(6,5)-enriched SWNT complex oxidized with the H_2_O_2_, K_2_IrCl_6_, or KMnO_4_, and then reduced with the catechin solution.

As shown in Figure 2a, when the H_2_O_2_ with a concentration of 9.8 mM was added to the dsDNA-(6,5)-enriched SWNT complex, the absorbance decreased by only 0.13%, relative to the initial state. When the catechin solution was added, the absorbance further decreased by 0.73%. Even when the H_2_O_2_ concentration was increased to 98 mM, the changes due to the addition of the H_2_O_2_ and catechin were slight (0.72% and 1.27%, respectively; see Appendix A). This result suggests that the H_2_O_2_ concentration does not significantly impact the absorbance of the complex.

Figure 2b shows the absorption spectra when K_2_IrCl_6_ with a concentration of 2.0 µM was added. The absorbance decreased by 23.3% with the addition of the K_2_IrCl_6_ and recovered by 31.0% with the addition of the catechin solution. To investigate the detectable range of the K_2_IrCl_6_ in the dsDNA-(6,5)-enriched SWNT complex, the absorbance was measured at different concentrations. The detectable range was found to be 0.5–5.0 µM.

Figure 2c shows the absorption spectra when the KMnO_4_ was added at a concentration of 0.5 µM. The absorbance decreased by 23.9% with the addition of the KMnO_4_ and recovered by 30.7% with the addition of the catechin solution. The detectable range of KMnO_4_ in the dsDNA-(6,5)-enriched SWNT complex was found to be 0.05–10.0 µM.

Figure 3a–c show examples of the changes in the NIR-absorbance spectra for the dsDNA-HiPco SWNT complex oxidized with the H_2_O_2_, K_2_IrCl_6_, or KMnO_4_ and reduced with the catechin solution.

HiPco SWNTs contain more than 10 types of chirality. Among them, we focused on the (6,5) chirality with an absorbance peak near 990 nm, the (8,7) chirality with a peak near 1130 nm, and the (9,4) chirality with a peak near 1265 nm [20].

Focusing on the (6,5) chirality in Figure 3a, when the H_2_O_2_ with a concentration of 9.8 mM was added to the dsDNA-HiPco SWNT complex, the absorbance decreased by only 2.3%. After adding the catechin solution, the absorbance slightly recovered to −1.0% from the initial state. This result is similar to the behavior of the (6,5) chirality of the dsDNA-(6,5)-enriched SWNT complex. On the other hand, the (8,7) and (9,4) chiralities showed a remarkable magnitude of change in absorbance compared to the (6,5) chirality. This result has been reported in a previous study [40]. The absorbance of the (8,7) chirality decreased by 15.0% with the addition of the H_2_O_2_ and recovered by 21.7% with the addition of the catechin solution. The detectable range of the H_2_O_2_ in the (8,7) chirality was 49 µM to 98 mM. The absorbance of the (9,4) chirality decreased by 7.40% with the addition of the H_2_O_2_ and recovered by 7.07% with the addition of the catechin solution. The detectable range of the H_2_O_2_ for the (9,4) chirality was the same as that of the (8,7) chirality. The difference in the magnitude of the change in absorbance of the H_2_O_2_ indicates that the (8,7) chirality is more sensitive than the (9,4) chirality.

Figure 3b shows the absorption spectra when the K_2_IrCl_6_ with a concentration of 1.0 µM was added. The absorbance of the (6,5) chirality decreased by 2.12% with the addition of the K_2_IrCl_6_ and recovered by 0.39% with the addition of the catechin solution. The detectable range of the K_2_IrCl_6_ with the (6,5) chirality was 0.1–10.0 µM. The absorbance of the (8,7) chirality decreased by 9.2% by adding the K_2_IrCl_6_ and recovered by 15.3% with the addition of the catechin solution. The detectable range of the K_2_IrCl_6_ with the (8,7) chirality was 0.1–1.5 µM. The absorbance of the (9,4) chirality decreased by 4.18% by adding the K_2_IrCl_6_ and recovered by 4.07% with the addition of the catechin solution. The detectable range of the K_2_IrCl_6_ with the (9,4) chirality was 0.1–2.0 µM.

Figure 3c shows the absorption spectra when the KMnO_4_ with a concentration of 0.1 µM was added. The absorbance of the (6,5) chirality decreased by 1.3% with the addition of the KMnO_4_ and recovered by 0.9% with the addition of the catechin solution. The detectable range of the KMnO_4_ in the (6,5) chirality was 0.025–1.0 µM. The absorbance of the (8,7) chirality decreased by 14.7% after adding the KMnO_4_ and recovered by 24.6% with the addition of the catechin solution. The detectable range of the KMnO_4_ in the (8,7) chirality was 0.025–0.1 µM. The absorbance of the (9,4) chirality decreased by 15.7% by adding the K_2_IrCl_6_ and recovered by 17.7% with the addition of the catechin solution. The detectable range of the KMnO_4_ in the (9,4) chirality was 0.025–0.25 µM.

Figure 4a–c show the change in the absorbance of the (6,5) chirality of the dsDNA-(6,5)-enriched SWNT complex with respect to changes in the concentrations of the H_2_O_2_, K_2_IrCl_6_, or KMnO_4_, respectively.

As shown in Figure 4a, no significant spectral change was observed when the H_2_O_2_ was added at the final concentrations of 98 and 9.8 mM. Figure 4b shows that a decrease in the absorbance corresponding to the change in the K_2_IrCl_6_ concentration was detected in the concentration range of 0.5–5.0 µM. Similarly, Figure 4c shows that a decrease in absorbance corresponding to changes in the KMnO_4_ concentration was detected in the concentration range of 0.05–10.0 µM. The absorbance change and wavelength peak shift of the dsDNA-(6,5)-enriched SWNT complex during oxidation and reduction are shown in Appendix A.

Figure 5a–c show the change in the absorbance of the (6,5) chirality of the dsDNA-HiPco SWNT complex with respect to changes in the concentration of the H_2_O_2_, K_2_IrCl_6_, or KMnO_4_, respectively.

As shown in Figure 5a, the absorbance decreased by up to 3.0% when the H_2_O_2_ concentration was 98 µM, but no significant difference was observed in the range of 49 µM to 98 mM. Figure 5b shows that a decrease in the absorbance corresponding to the change in the K_2_IrCl_6_ concentration was detected in the range of 0.1–5.0 µM. Figure 5c indicates that a decrease in the absorbance corresponding to the KMnO_4_ concentration was detected in the range of 0.025–1.0 µM. The absorbance change and wavelength peak shift of the (6,5) chirality of the dsDNA-HiPco SWNT complex during oxidation and reduction are shown in Appendix A.

Figure 6a–c show the change in the absorbance of the (8,7) chirality of the dsDNA-HiPco SWNT complex with respect to the change in the concentration of the H_2_O_2_, K_2_IrCl_6_, or KMnO_4_, respectively.

As shown in Figure 6a, when the H_2_O_2_ concentration changed from 49 µM to 9.8 mM, the magnitude of change in the absorbance increased. However, the magnitude decreased at 98 mM. Figure 6b shows a decrease in absorbance corresponding to an increase in the K_2_IrCl_6_ concentration in the range of 0.1–1.5 µM. Figure 6c indicates a decrease in the absorbance corresponding to an increase in the KMnO_4_ concentration in the range of 0.025–0.1 µM. The absorbance change and wavelength peak shift of the (8,7) chirality of the dsDNA-HiPco SWNT complex during oxidation and reduction are shown in Appendix A.

Figure 7a–c show the change in the absorbance of the (9,4) chirality of the dsDNA-HiPco SWNT complex with respect to the change in the concentration of the H_2_O_2_, K_2_IrCl_6_, or KMnO_4_, respectively.

As shown in Figure 7a, when the H_2_O_2_ concentration changed from 49 µM to 9.8 mM, the magnitude of the change in absorbance increased. However, the magnitude decreased at 98 mM. Figure 7b shows the decrease in the absorbance corresponding to an increase in the K_2_IrCl_6_ concentration in the range of 0.1–2.0 µM. Figure 7c indicates a decrease in the absorbance corresponding to an increase in the KMnO_4_ concentration in the range of 0.025–0.25 µM. The absorbance change and wavelength peak shift of the (9,4) chirality of the dsDNA-HiPco SWNT complex during oxidation and reduction are shown in Appendix A.

## 3. Discussion

We found that changes in the H_2_O_2_ concentration from 49 µM to 98 mM did not cause a dramatic difference in the absorbance for any chirality. The (6,5) chirality in both the (6,5)-enriched SWNT and HiPco SWNT complexes was less sensitive to the H_2_O_2_, which has a low oxidizing power. Both the (8,7) and (9,4) chiralities responded sensitively to the H_2_O_2_. When the H_2_O_2_ concentration was 9.8 mM, the magnitude of the change in absorbance was at a maximum for both chiralities, though the (8,7) chirality was more sensitive than the (9,4) chirality. Because the (6,5) chirality was less sensitive to oxidation, it was possible to detect oxidation over a wide range of concentrations for the K_2_IrCl_6_ and KMnO_4_, which contain atoms with high oxidizing powers. On the other hand, because the (8,7) and (9,4) chiralities were highly sensitive to oxidation, the peaks of the absorption spectra did not appear at high concentrations, and the detectable range narrowed. Moreover, the detectable range of the (8,7) chirality was narrower than that of the (9,4) chirality, as the (8,7) chirality was more sensitive toward the K_2_IrCl_6_ and KMnO_4_.

Here, we focused on the oxidizing power, considering that oxidation reflects the movement of electrons. The oxidizing power is the total number of electrons that an atom either gains or loses to form a chemical bond with another atom. Each atom that participates in an oxidation-reduction reaction is assigned an oxidizing power that reflects its ability to acquire, donate, or share electrons. When an atom is in an oxidized state, the oxidizing power becomes positive, and a larger value indicates a more electron-deficient atom. On the contrary, when the atom is in a reduced state, it assumes a negative oxidizing power, and a larger value indicates a more electron-rich atom. The oxidizing powers of the oxidizing agents used in this study were −1 for the H_2_O_2_, +4 for the Ir of K_2_IrCl_6_, and +7 for the Mn of KMnO_4_. 

First, we compared the oxidizing power. Table 1 shows the relationship between the oxidizing power and the magnitude of the change in absorbance for each chirality in the dsDNA-SWNT complexes. We compared the concentrations of the oxidants required to achieve relatively similar changes in the absorbance for each chirality. For example, the H_2_O_2_ (oxidizing power of −1) at a concentration of 9.8 mM reduced the absorbance in the (6,5) chirality of the (6,5)-enriched SWNT by 0.13%, while the K_2_IrCl_6_ (oxidizing power of +4) reduced it by 0.10% at a concentration of 0.5 µM. Both oxidizing agents reduced the absorbance by approximately 0.1%, but the concentration of the H_2_O_2_ was 1.9 × 10^4^ times higher than that of the K_2_IrCl_6_. KMnO_4_ (oxidizing power of +7) reduced the absorbance by 1.60% at a concentration of 0.05 µM. This magnitude of change was larger than that of the H_2_O_2_, but no smaller change could be detected. Therefore, by comparing the absorbance change in KMnO_4_ at 0.05 µM, the concentration of the H_2_O_2_ was 1.9 × 10^5^ times higher. The concentration ratios of the oxidants for the other chiralities are shown in Table 1. These results suggest that the difference in oxidizing power has a significant effect on the absorbance of the SWNT chirality.

Next, we compared the sensitivity of the chirality to the concentration of the oxidant. Figure 8a–c show the absorbance changes for each chirality when the same concentration of oxidizing agent was added.

Neither the (6,5) chirality of (6,5)-enriched SWNT complex nor the (6,5) chirality of the HiPco SWNT complex reacted much in 9.8 mM H_2_O_2_, while the (8,7) and (9,4) chiralities responded sensitively. In the case of the 1.0 µM K_2_IrCl_6_, the (6,5) chirality of the (6,5)-enriched SWNT complex responded with similar sensitivity as the (8,7) chirality. In the case of the KMnO_4_, the (8,7) and (9,4) chiralities reacted sensitively, even at a low concentration of 0.1 µM. These results suggest that the change in the absorbance differs depending on the chirality, even at the same oxidant concentration.

Finally, we examined the concentration range of the oxidants that each chirality could detect. As shown in Appendix A, the (6,5) chirality of (6,5)-enriched SWNT complex and the (6,5) chirality of HiPco SWNT complex showed no significant change in the absorbance, even as the concentration of the H_2_O_2_ increased, thus they are not discussed here. As shown in Appendix A, the concentration range of the H_2_O_2_ that could be detected by both the (8,7) and (9,4) chiralities was 49 µM to 98 mM.

The detectable ranges of the K_2_IrCl_6_ for the (6,5) chirality of the (6,5)-enriched SWNT complex and the (6,5), (8,7), and (9,4) chiralities of the HiPco SWNT complex are shown in Appendix A, respectively. The recovery of the absorbance with the addition of the catechin solution is also shown in the same figures. To specify the detectable range of the K_2_IrCl_6_, the peak of the absorption spectra was identified while increasing the concentration of the oxidant (Appendix A).

The detectable ranges of the KMnO_4_ for the (6,5) chirality of the (6,5)-enriched SWNT complex and the (6,5), (8,7), and (9,4) chiralities of the HiPco SWNT complex, are shown in Appendix A, respectively. The recovery of the absorbance with the addition of the catechin solution is also shown in the same figures. To specify the detectable range of the KMnO_4_, the peak of the absorption spectra was identified while increasing the concentration of the oxidant (Appendix A).

Figure 9a,b show the detectable range of the oxidant concentration for each chirality revealed in this study.

Table 2 shows the peak shift in the absorption spectra of each chirality for the different concentrations of oxidants.

For the (6,5) chirality of the (6,5)-enriched SWNT complex, no peak shift was observed when the H_2_O_2_ was added. No significant peak shift was observed at low concentrations of K_2_IrCl_6_ up to 2.0 µM, but a peak shift toward the long-wavelength region was observed at 5.0 µM. As the KMnO_4_ concentration increased, the peak shift toward the short-wavelength region increased. The (6,5) chirality of the HiPco SWNT complex did not show a significant peak shift for any oxidant, while the (8,7) chirality showed a significant peak shift in proportion to the concentration of any oxidizing agent. This suggests that the (8,7) chirality is sensitive to any oxidizing agent. The (9,4) chirality did not show a significant peak shift for any oxidant.

## 4. Materials and Methods

SWNTs manufactured using the high-pressure carbon monoxide (HiPco) synthesis method were purchased from Raymor Industries Inc. (Boisbriand, QC, Canada). (6,5)-enriched SWNT powder (No. 773735-250G), produced by the CoMoCAT synthesis method, and dsDNA (deoxyribonucleic acid sodium salt from salmon testes, D1626) were purchased from Sigma-Aldrich Co. LLC (St. Louis, MO, USA). H_2_O_2_ (approx. 30%, 084-07441), catechin (553-74471), and K_2_IrCl_6_ (99%, 77-6500) were purchased from FUJIFILM Wako Pure Chemical Corporation (Osaka, Japan). KMnO_4_ solution (42000375) was obtained from Hayashi Pure Chemical Ind. LTD. (Osaka, Japan). 

A 1 mg/mL dsDNA solution was prepared with 10 mM tris (hydroxymethyl) aminomethane-HCl (tris-HCL) buffer (pH 7.9). To untangle the dsDNA molecules, the solution was sonicated on ice in a bath-type ultrasonicator (80 W) for 90 min. Finally, the dsDNA solution was gently shaken for 3 h. To prepare the dsDNA-HiPco SWNT complex, 0.5 mg of the SWNT powder and 1 mL of the dsDNA stock solution were mixed and sonicated on ice for 1.5 h using a probe-type sonicator (3 W, VCX130, Sonic & Materials, Inc., Newtown, CT, USA). The supernatant of the prepared dsDNA-HiPco SWNT dispersion was extracted by centrifuging at 17,360× *g* for 3 h at 8 °C and then stored [41,42,43].

Thereafter, 0.5 mg of the (6,5)-enriched SWNT powder was suspended in 1 mL of the dsDNA solution. The samples were sonicated using a probe-type sonicator (3 W) for 1.5 h, followed by centrifugation at 17,360× *g* for 3 h at 8 °C. The supernatant was collected as a dsDNA-(6,5)-enriched SWNT suspension.

A near-infrared (NIR) spectrometer (SolidSpec-3700DUV, Shimadzu Corporation, Kyoto, Japan) was employed for absorbance measurements (700–1350 nm). For the NIR measurements, 100 μL of the dsDNA-HiPco SWNT suspension and 880 μL of the tris-HCL buffer solution were mixed in a cuvette, and the initial spectra were recorded. Subsequently, 10 μL of the H_2_O_2_ diluted with sterilized water was added to the samples, followed by incubation for 30 min at 21 °C. Similarly, 10 μL of one of two oxidants (K_2_IrCl_6_ or KMnO_4_) diluted with sterilized water was added to the samples, followed by incubation for 10 min at 21 °C. The spectra of the mixed samples were then measured. Finally, 10 μL of the catechin solution (final concentration 0.15 μg/mL) was added to the samples, and the spectra were measured again after incubation for 10 min at 21 °C.

For the dsDNA-(6,5)-enriched SWNT suspensions, an ultraviolet-visible spectrophotometer (V-630, Jasco Corporation, Hachioji City, Tokyo, Japan) was employed for the NIR spectra measurements (700–1350 nm). For this, 50 μL of the dsDNA-(6,5)-enriched SWNT complex and 440 μL of a buffer solution (pH 7.9) were mixed in a cuvette, and the initial spectra were recorded. Subsequently, 5 μL of the H_2_O_2_ diluted with sterile water was added to the samples and incubated for 30 min at 21 °C. Similarly, 5 μL of one of two oxidants (K_2_IrCl_6_ or KMnO_4_) diluted with sterilized water was added to the samples followed by incubation for 10 min at 21 °C. The spectra of the samples were then measured. Finally, 5 μL of the catechin solution (final concentration 0.15 μg/mL) was added to the samples, and the spectra were measured again after 10 min of incubation at 21 °C.

The final concentration of the oxidants was changed stepwise with H_2_O_2_, K_2_IrCl_6_, or KMnO_4_ in the range of 49 µM to 19.6 mM, 0.1–10 µM, and 0.025–2.0 µM, respectively, and the spectra were detected at each step. Triplicate NIR measurements for each experiment were recorded to verify the reproducibility.

## 5. Conclusions

This research suggests that a difference in the oxidizing power has a significant effect on the absorbance of an SWNT chirality. The magnitude of the H_2_O_2_ concentration did not significantly affect the absorbance for any chirality. The (6,5) chirality of both the dsDNA-(6,5)-enriched SWNT and dsDNA-HiPco SWNT complexes are slightly sensitive to H_2_O_2_, but it was possible to detect high concentrations of K_2_IrCl_6_ and KMnO_4_.

The detectable range of the H_2_O_2_ was the same for both the (8,7) and (9,4) chiralities of the dsDNA-HiPco SWNT complex. However, the magnitude of the absorbance change in the H_2_O_2_ indicates that the (8,7) chirality was more sensitive than the (9,4) chirality. The (8,7) chirality was also more sensitive than the (9,4) chirality to the concentrations of the K_2_IrCl_6_ and KMnO_4_, so the detectable range was narrower than that of the (9,4) chirality. Therefore, we determined that the magnitude of the change in absorbance differs depending on the chirality, even at the same oxidant concentration.

## Figures and Tables

**Figure 1 molecules-26-01091-f001:**
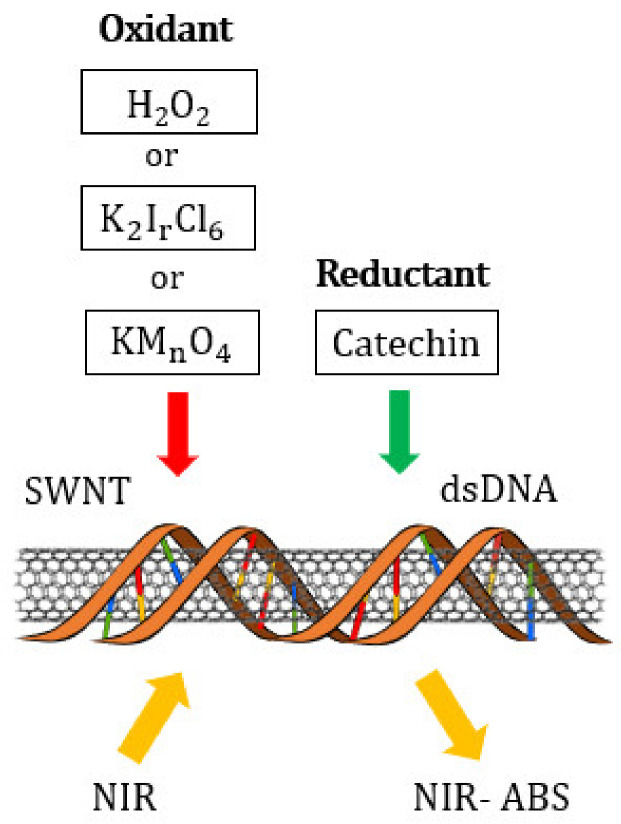
Conceptual diagram of experiment.

**Figure 2 molecules-26-01091-f002:**
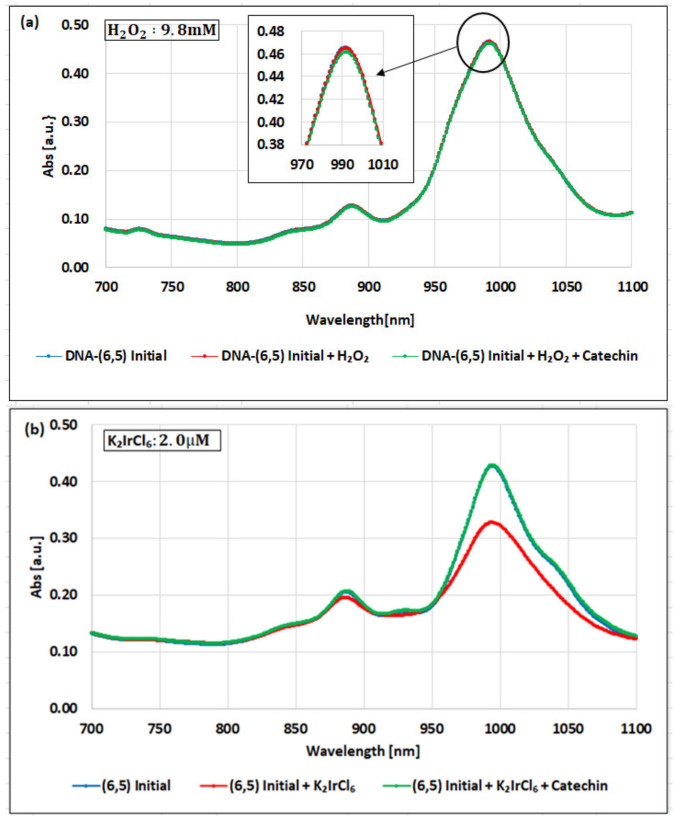
NIR-absorbance spectra of the dsDNA-(6,5)-enriched SWNT complex during redox with (**a**) H_2_O_2_ and catechin, (**b**) K_2_IrCl_6_ and catechin, and (**c**) KMnO_4_ and catechin. The data are presented as the average of three independent experiments.

**Figure 3 molecules-26-01091-f003:**
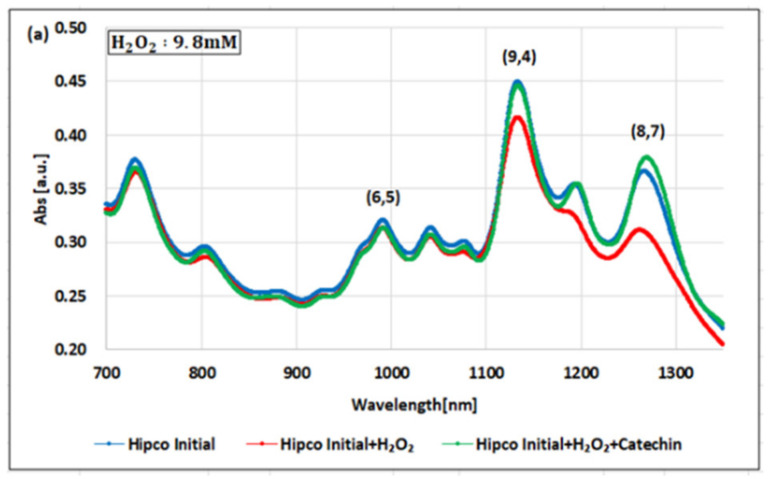
NIR-absorbance spectra of the dsDNA-HiPco SWNT complex during redox with the (**a**) H_2_O_2_ and catechin, (**b**) K_2_IrCl_6_ and catechin, and (**c**) KMnO_4_ and catechin. The data are presented as the average of three independent experiments.

**Figure 4 molecules-26-01091-f004:**
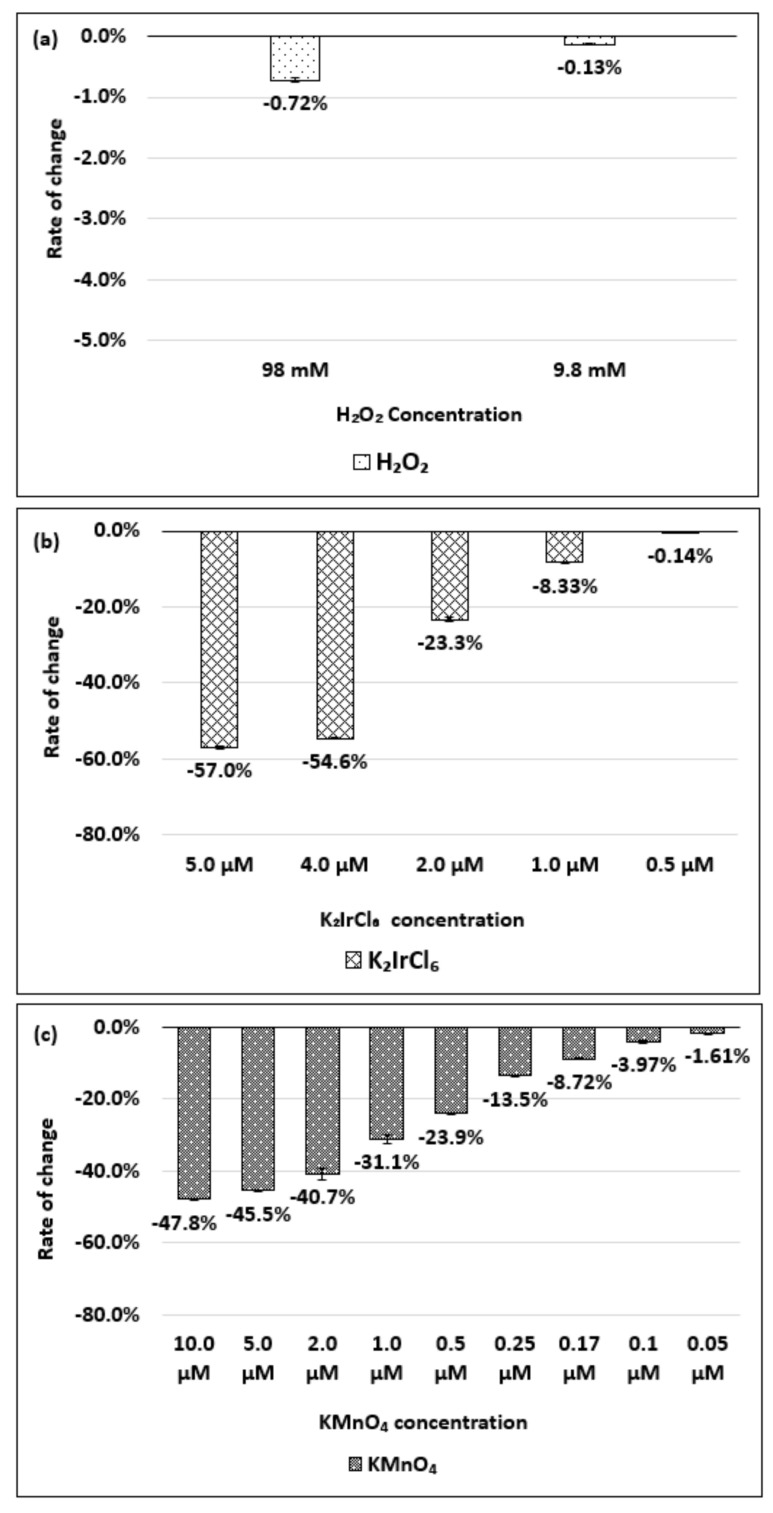
Absorbance change for the dsDNA-(6,5)-enriched SWNT complex, relative to the initial state, based on the concentration of (**a**) H_2_O_2_, (**b**) K_2_IrCl_6_, and (**c**) KMnO_4_ added. The data are presented as the average of three independent experiments.

**Figure 5 molecules-26-01091-f005:**
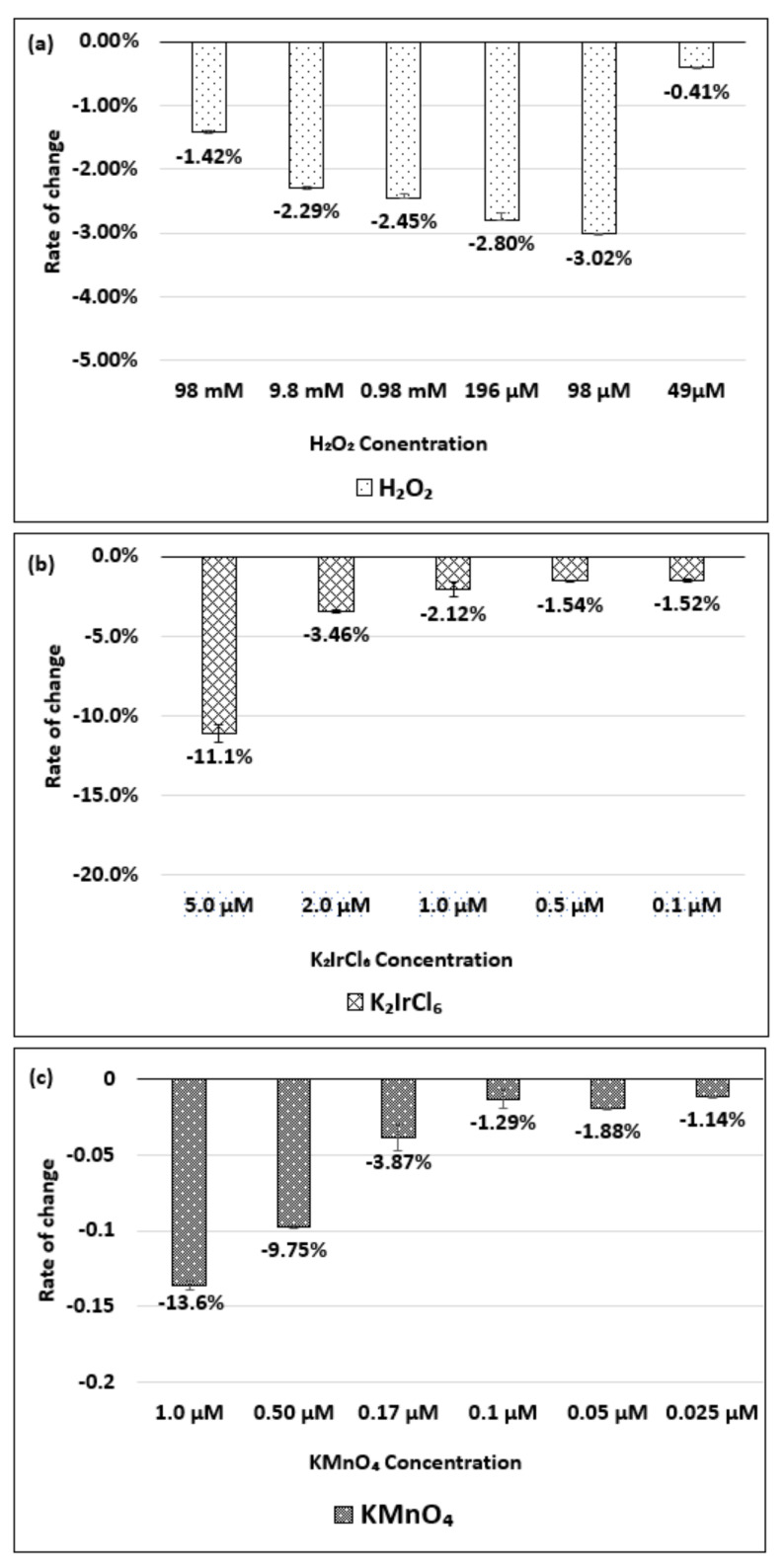
Absorbance change for the (6,5) chirality of the dsDNA-HiPco SWNT complex, relative to the initial state, based on the concentration of the (**a**) H_2_O_2_, (**b**) K_2_IrCl_6_, and (**c**) KMnO_4_. The data are presented as the average of three independent experiments.

**Figure 6 molecules-26-01091-f006:**
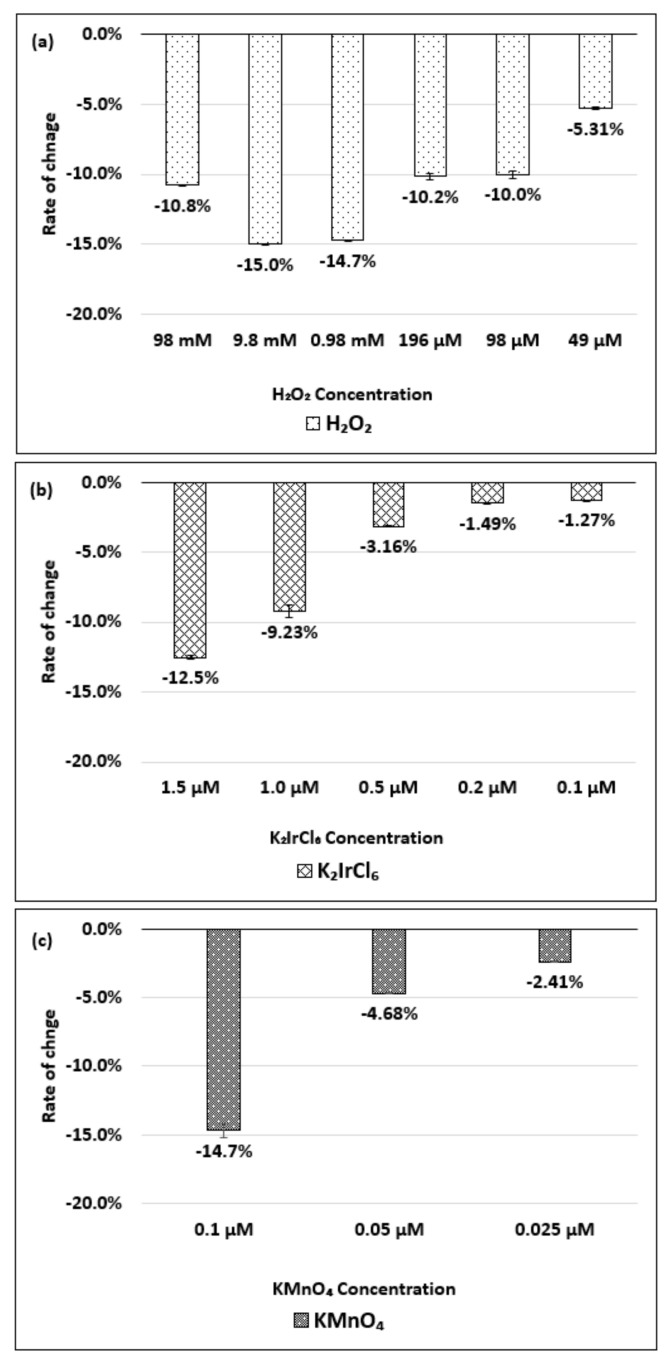
Absorbance change for the (8,7) chirality of the dsDNA-HiPco SWNT complex, relative to the initial state, based on the concentration of the (**a**) H_2_O_2_, (**b**) K_2_IrCl_6_, and (**c**) KMnO_4_. The data are presented as the average of three independent experiments.

**Figure 7 molecules-26-01091-f007:**
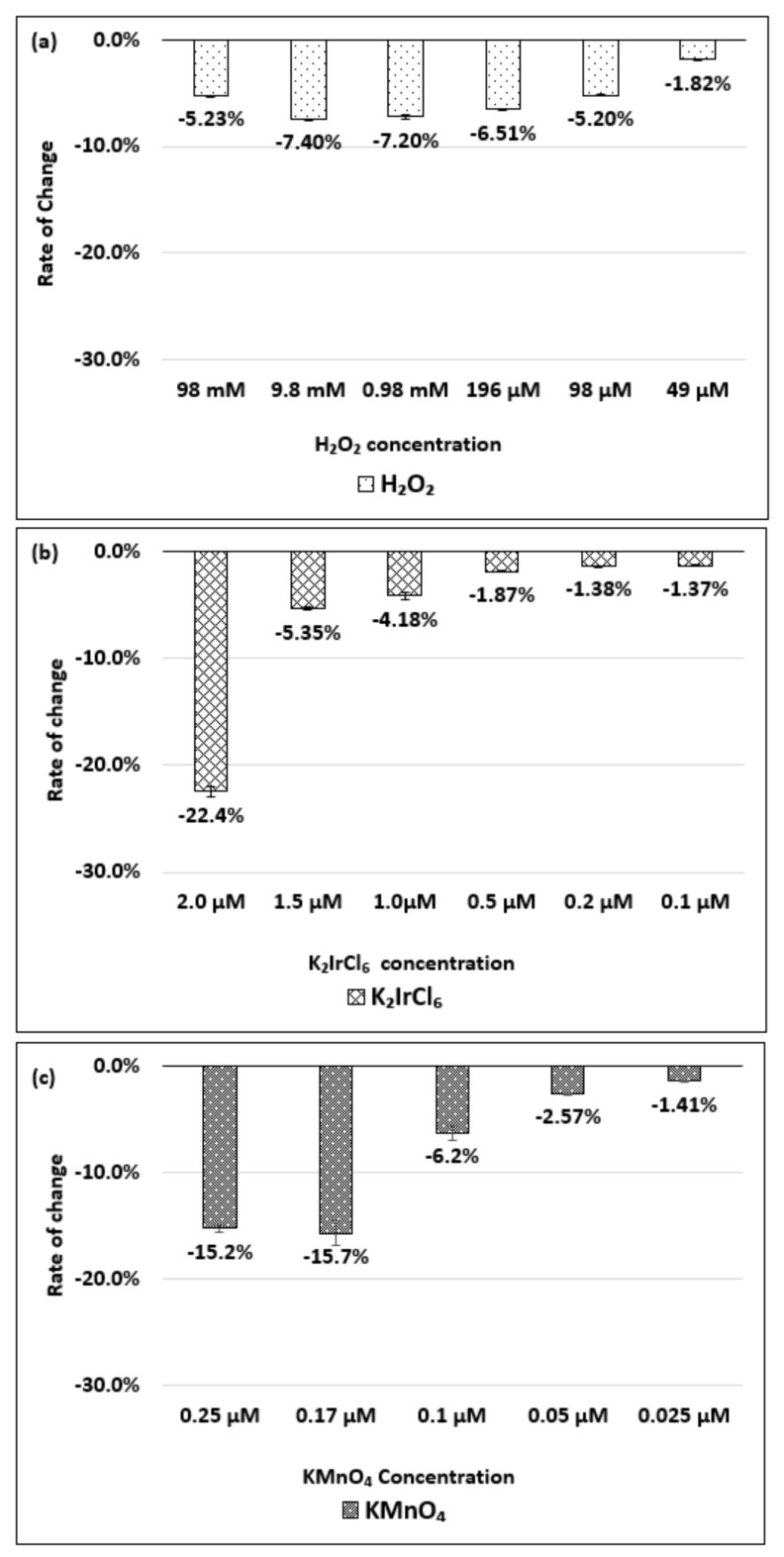
Absorbance change for the (9,4) chirality of the dsDNA-HiPco SWNT complex, relative to the initial state, based on the concentration of the (**a**) H_2_O_2_, (**b**) K_2_IrCl_6_, and (**c**) KMnO_4_. The data are presented as the average of three independent experiments.

**Figure 8 molecules-26-01091-f008:**
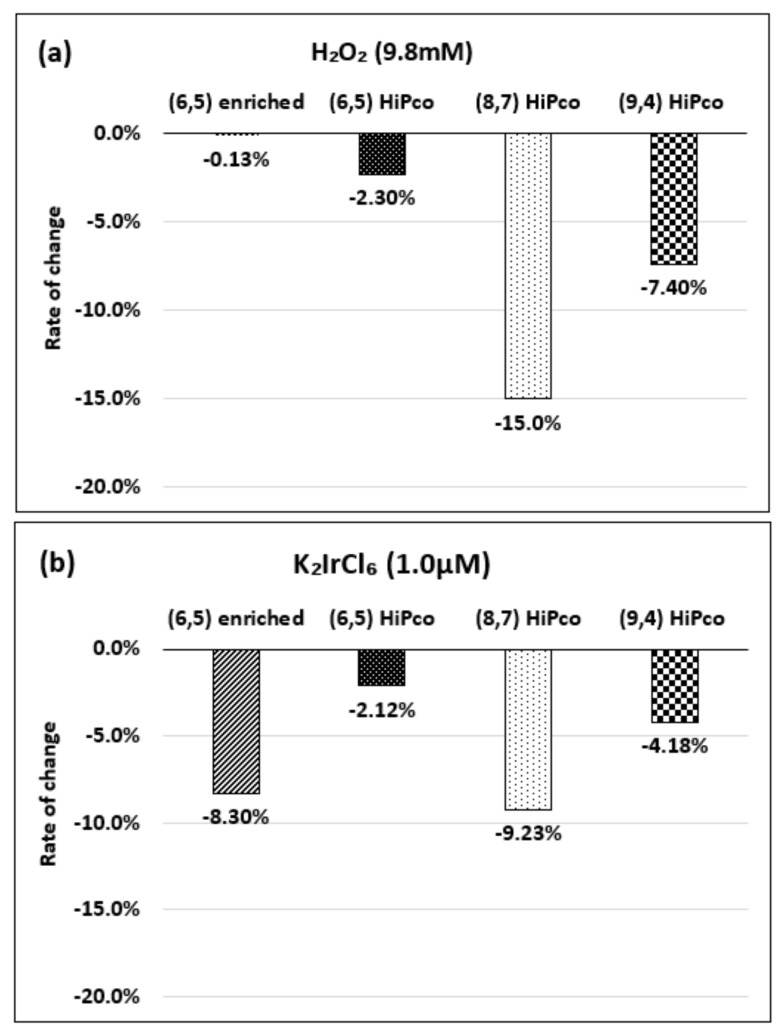
Change in the absorbance for each chirality with the addition of (**a**) 9.8 mM H_2_O_2_, (**b**) 1.0 µM K_2_IrCl_6_, and (**c**) 0.1 µM KMnO_4_. The data are presented as the average of three independent experiments.

**Figure 9 molecules-26-01091-f009:**
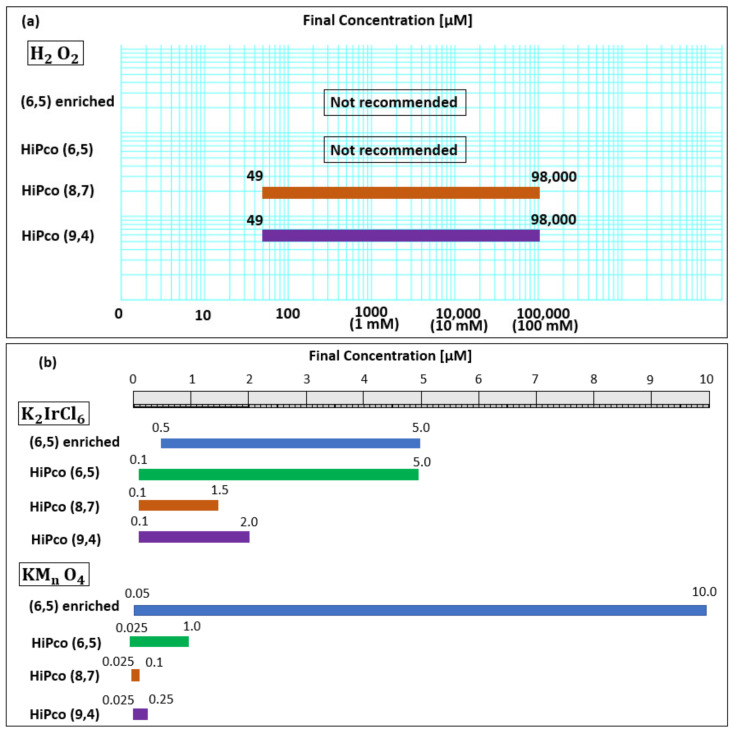
Detectable concentration ranges of (**a**) H_2_O_2_ and (**b**) K_2_IrCl_6_ and KMnO_4_ for each chirality. The blue line shows the detectable range of the (6,5) chirality of (6,5) -enriched SWNT complex. The green line, brown line, and purple line indicate the detectable range of the (6,5), (8,7), or (9,4) chirality of the HiPco SWNT complex, respectively.

**Table 1 molecules-26-01091-t001:** Relationship between the oxidizing power and absorbance change for each chirality. The upper part of the chirality column shows the concentration of the added oxidant, and the lower part shows the change in absorbance at that time, relative to the initial state. The data are presented as the average of three independent experiments.

Oxidizing Power Chirality	−1 (H_2_O_2_)	+4 (K_2_IrCl_6_)	+7 (KMnO_4_)	Concentration Ratio of Oxidizing Power		
				H_2_O_2_/K_2_IrCl_6_	H_2_O_2_/KMnO_4_	K_2_IrCl_6_/KMnO_4_
(6,5) (6,5)-enriched	9800 µM	0.5 µM	0.05 µM	1.9 × 10^4^	1.9 × 10^5^	10
Absorbance Change	−0.13%	−0.10%	−1.60%	-	-	-
(6,5) HiPco	9800 µM	1.0 µM	0.05 µM	0.98 × 10^4^	1.9 × 10^5^	20
Absorbance Change	−2.30%	−2.12%	−1.90%	-	-	-
(8,7) HiPco	980 µM	1.50 µM	0.10 µM	6.5 × 10^2^	0.98 × 10^4^	15
Absorbance Change	−14.7%	−12.5%	−14.7%	-	-	-
(9,4) HiPco	98 µM	1.50 µM	0.05 µM	6.5 × 10	1.9×10^3^	30
Absorbance Change	−5.20%	−5.35%	−6.20%	-	-	-

**Table 2 molecules-26-01091-t002:** Absorption spectra peak shift for each chirality by the oxidant concentration. The data are presented as the average of three independent experiments.

Oxidant Chirality	H_2_O_2_ Concentration						K_2_IrCl_6_ Concentration				KMnO_4_ Concentration				
	49 µM	98 µM	196 µM	0.98 mM	0.98 mM	98 mM	0.5 µM	1.0 µM	2.0 µM	5.0 µM	0.1 µM	0.25 µM	0.5 µM	1.0 µM	5.0 µM
(6,5) enriched	-	-	-	-	0.0	0.0	−0.3	−0.3	0.0	5.7	−0.07	−2.0	−4.0	−6.3	−16.7
(6,5) HiPco	0.0	0.0	0.3	−0.3	−0.2	0.0	0.0	0.0	−0.5	0.0	−0.5	0.0	0.0	2.7	-
(8,7) HiPco	−1.5	−2.7	−2.7	−4.8	−4.7	−4.5	−0.8	−3.7	−5.7	-	−6.2	-	-	-	-
(9,4) HiPco	0.0	0.0	1.0	0.0	−0.2	−0.8	0.0	0.0	−2.0	-	−0.5	−1.5	-	-	-

## Data Availability

The data presented in this study are available in the article or the Appendix A.

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
