# Peer review of "Optical Response Characteristics of Single-Walled Carbon Nanotube Chirality Exposed to Oxidants with Different Oxidizing Power"

_molecules, 2021, doi:10.3390/molecules26041091_

Round 1
Reviewer 1 Report
In this manuscript authors investigated the relationship between the chirality of single-walled carbon nanotubes (SWNTs) and the oxidation number of oxidants by measuring the near-infrared (NIR) absorption spectra of two double-stranded DNA-SWNT complexes with the addition of three oxidants (H2O2, K2IrCl6 and KMnO4) with different oxidation numbers. In general, research work is well done and presented. I recommend its publication in Molecules.
Reviewer 2 Report
The manuscript of Matsukawa and Umemura describes the results of a study on the relationship between the three-dimensional structure (chirality) of SWNTs and the oxidation numbers of three oxidants (hydrogen peroxide, potassium hexachloroiridate, and potassium permanganate) by measuring the near-infrared absorption spectra of two double-stranded DNA-SWNT complexes after the addition of the oxidants. The manuscript is well-written, the study has been designed and executed properly. On this basis, in my opinion the manuscript deserves to be published in Pharmaceuticals after a minor revision.
Comments and suggestions:
1) Abstract, line 10: please change "geometric" to "three-dimensional";
2) Introduction: for clarity, I suggest to the Authors to describe, in the Introduction section, the structural features of the SWNTs, the meaning of chiral vectors, chiral index and the general features of the chirality of SWNTs. In this regards, a proper reference has to be introduced. I suggest: Lu-Chang Qin, Determination of the chiral indices (n,m) of carbon nanotubes by electron diffraction Phys. Chem. Chem. Phys., 2007, 9, 31–48. Moreover, the recent paper of the authors “Chirality luminescent properties of single-walled carbon nanotubes during redox reactions”, Optical Materials 112 (2021) 110748 should be introduced in the Introduction;
3) please check the manuscript for typos.
Reviewer 3 Report
Matsukawa and Umemura have studied by NIR-absorption the behavior of two types of DNA-SWNT complexes upon addition of different oxidants at different concentrations. The work was well conducted and clearly presented. It is likely that this work attracts much interest in the vast community of people working in the field of carbon nanotubes. I recommend acceptance in Molecules only after the authors have modified their manuscript according to the comments given below:
1) My major concern is regarding the term “oxidation number” and the ranking the authors give based only on this parameter. The ability to gain or lose an electron is different from an atom to another. The three oxidants used in this work are based on three different elements, namely O, Ir and Mn, and cannot be ranked just on the basis on their oxidation number. Therefore, I suggest to use the terms “oxidizing power” or “oxidizing ability” to compare the different oxidants. It does not change the content of the article since KMnO4 is more oxidizing than K2PtCl6 which is more oxidizing than H2O2.
2) Please rewrite correctly the reference section by following the guidelines
